# Comprehensive Immune Profiling Unveils a Subset of Leiomyosarcoma with “Hot” Tumor Immune Microenvironment

**DOI:** 10.3390/cancers15143705

**Published:** 2023-07-21

**Authors:** Xiaolan Feng, Laurie Tonon, Haocheng Li, Elodie Darbo, Erin Pleasance, Nicolas Macagno, Armelle Dufresne, Mehdi Brahmi, Julien Bollard, Francoise Ducimetière, Marie Karanian, Alexandra Meurgey, Gaëlle Pérot, Thibaud Valentin, Frédéric Chibon, Jean-Yves Blay

**Affiliations:** 1Tom Baker Cancer Center, Department of Medical Oncology, University of Calgary, Calgary, AB T2N 4N2, Canada; 2Synergie Lyon Cancer, Gille Thomas Bioinformatice Platform, Centre Léon Bérard, 69008 Lyon, France; 3Department of Mathematics and Statistics, University of Calgary, Calgary, AB T2N 4N1, Canada; 4BRIC, INSERM U1312, Université de Bordeaux, 33600 Bordeaux, France; 5Canada’s Michael Smith Genome Sciences Centre, BC Cancer, Vancouver, BC V5Z 4S6, Canada; 6Department of Pathology, Aix Marseille University, INSERM, APHM MMG, UMR1251, Marmara Institute, La Timone Hospital, 13005 Marseille, France; 7Department of Medical Oncology, Centre Leon Bérard, 69008 Lyon, France; 8Centre Léon Bérard, Department of Pathology, 69008 Lyon, France; 9Department of Pathology, Institut Claudius Régaud, IUCT-Oncopole, 31000 Toulouse, France; 10Department of Medical Oncology, Centre Léon Bérard, University Claude Bernard Lyon, 69008 Lyon, France; jean-yves.blay@lyon.unicancer.fr

**Keywords:** whole transcriptomic profiling (WTP), leiomyosarcoma (LMS), tumor immune microenvironment (TIME), immune checkpoint inhibitors (ICIs)

## Abstract

**Simple Summary:**

Leiomyosarcoma (LMS) is thought to be an immune cold tumor that generally does not respond to immune checkpoint inhibitors (ICIs). To date, there is no validated immune biomarker used in LMS patients. The tertiary lymphoid structure (TLS) is the only potential predictive biomarker, but rarely present in LMS. Our study is the first study to investigate the immune biomarker using comprehensive transcriptomic profiling solely focused on tumor immune microenvironment (TIME) in LMS. Our study identified a subset of LMS with an active (“hot”) tumor immune microenvironment (TIME) that is consistently associated with several immune signatures validated in other cancers in the clinical setting. Our study supports the further development of TIME multi-gene immune signature predictive biomarker that can be embedded in the future prospective clinical trials to evaluate its clinical utility to select LMS patients for ICIs.

**Abstract:**

**Purpose:** To investigate the immune biomarker in Leiomyosarcoma (LMS), which is rare and recognized as an immune cold cancer showing a poor response rate (<10%) to immune checkpoint inhibitors (ICIs). However, durable response and clinical benefit to ICIs has been observed in a few cases of LMS, including, but not only, LMS with tertiary lymphoid structure (TLS) structures. **Patients and methods:** We used comprehensive transcriptomic profiling and a deconvolution method extracted from RNA-sequencing gene expression data in two independent LMS cohorts, the International Cancer Genome Consortium (ICGC, N = 146) and The Cancer Genome Atlas (TCGA, N = 75), to explore tumor immune microenvironment (TIME) in LMS. **Results:** Unsupervised clustering analysis using the previously validated two methods, 90-gene signature and Cell-type Identification by Estimating Relative Subsets of RNA Transcripts (CIBERSORT), identified immune hot (I-H) and immune high (I-Hi) LMS, respectively, in the ICGC cohort. Similarly, immune active groups (T-H, T-Hi) were identified in the TCGA cohort using these two methods. These immune active (“hot”) clusters were significantly associated, but not completely overlapping, with several validated immune signatures such as sarcoma immune class (SIC) classification and TLS score, T cell inflamed signature (TIS) score, immune infiltration score (IIS), and macrophage score (M1/M2), with more patients identified by our clustering as potentially immune hot. **Conclusions:** Comprehensive immune profiling revealed a subset of LMS with a distinct active (“hot”) TIME, consistently associated with several validated immune signatures in other cancers. This suggests that the methodologies that we used in this study warrant further validation and development, which can potentially help refine our current immune biomarkers to select the right LMS patients for ICIs in clinical trials.

## 1. Background

Leiomyosarcoma (LMS) is a malignant mesenchymal tumor deriving smooth muscle differentiation with an estimated incidence of ~10% of all sarcomas [1,2]. This aggressive malignancy with propensity for systemic spread is associated with high recurrence rates and a poor overall survival of less than 18 months when it metastasizes [3].

According to tissue origin, subtypes are defined as soft tissue (STLMS) and uterine (uLMS). uLMS are often hormone receptors-positive cancer [4], and more often exhibit a distinct gene expression signature, a so-called “BRCAness”, resulting from homologous recombination deficiency (HRD) [5,6]. In addition, gene expression patterns led to classify LMS into conventional (c), uterogenic (ut) and inflammatory (i)-LMS, enriched in muscle-related transcripts, uterine-like gene, and immune markers subsets [7]. iLMS has the worse prognosis [7]. Whether the LMS classification (in particular iLMS) is associated with responses to immunotherapy remains to be clarified.

Although the first principles of cancer immunotherapy were evidenced in sarcoma more than a century ago [8], the numerous breakthroughs in immunotherapy, specifically through the use of immune checkpoint inhibitors (ICIs), achieved in the last decade drastically changed the therapeutic landscape and substantially improved the survival of cancer patients. However, ICIs show disappointing results in adult sarcoma. A recent systemic review and meta-analysis revealed that ICI led to only 14% of overall objective response rate (ORR) in sarcomas and to only 0–10% of ORR in LMS (both uterine and non-uterine) [9]. However, durable responses to ICIs have been reported [10,11] and real-world practice urges the need for predictive biomarker(s) to help appropriate the selection of patients with LMS eligible to ICIs. This is paramount in LMS considering the unfavorable “risk–benefit balance” in using ICIs, reporting low ORR in unselected patient population, occasional serious side effects, and substantial financial costs.

LMS belongs to sarcoma with complex karyotype associated with copy-number alterations (CNAs). In contrast to its counterpart sarcoma subtype driven by a pathognomonic genomic alteration (often translocations), higher PDL1 expression, immune infiltrates, and antigen presentation is observed [12]. For example, ~40–70% LMS express PDL1 [13,14,15], but no association with improved survival [14] nor ICIs response was reported [3,16]. Predictive biomarkers for ICIs include tumor mutational burden (TMB) and microsatellite instability (MSI). TMB is globally low (median TMB:1.5–2.5 mutations/Mb, with less than 1% harboring >20 mutations/Mb) in STS (with the only exception of cutaneous angiosarcoma related to UV exposure) [17]. In addition, MSI is generally extremely low (<1%), if not absent, in adult sarcoma [18,19]. Furthermore, compared with other CNAs-driven sarcoma such as undifferentiated pleomorphic sarcoma, LMS is rather poorly infiltrated by CD8 T cells and has high macrophage (M) M2/M1 (immune suppressive/immune promoting) ratio [20,21,22,23], which both characterized tumor immune microenvironment (TIME) with immune cold phenotype.

Most studies exploring prognostic and predictive immune biomarkers in adult sarcoma are limited in sample size (<100) and/or merge dissimilar histological subtypes therefore exposing limits in result interpretation, balancing statistical power and preservation of inherent specificities related to biological complexity and heterogeneity in sarcoma. Current knowledge faces conflicting results and high controversy in the field [24].

To date, the characterization of TIME revealed one single potential predictive immune biomarker, the sarcoma immune high subclass E (SIC E), sharing characteristics with tertiary lymphoid structure (TLS) as shown in the B cell lineage signature. SIC E predicted response to the ICIs pembrolizumab in adult STS in SARC028 study [25] and results were further confirmed prospectively in the PembroSarc study [26]. Three independent cohorts with various types of cancer treated with ICIs revealed that the presence of mature TLS was associated with better survival, regardless of CD8 T cell infiltrates or PDL1 expression [27]. This highlights the importance of improved TIME characterization to provide a potential predictive biomarker in selecting patients that are potentially better responders to immunotherapy. SIC E is currently used in several ongoing ICIs clinical trials in sarcoma (NCT02406781 and NCT04095208). According to this novel immune-based classification, LMS mainly belongs to SIC A (“immune desert”) or B (“immune low”) subclass with only a few showing TLS [25]. Therefore, LMS is considered as one of the “coldest” sarcoma histology subtypes and generally reported low ORR in clinical trials [9,28,29]. In addition, TLS, as immune biomarker, may not be sufficient to appropriately select LMS patients as candidates for ICIs. The PembroSarc study showed only one patient responder to pembrolizumab out of the six LMS patients with TLS-positive tumors [26].

We used comprehensive transcriptomic profiling and a deconvolution method extracted from RNA-sequencing gene expression data from two independent LMS cohorts, International Cancer Genome Consortium (ICGC), N = 146; and Cancer Genome Atlas (TCGA), N = 75) to explore the landscape of TIME in the single sarcoma subtype. LMS Immune profiling and clustering data were therefore associated with clinical factors, outcomes, and recently reported TIME signatures correlated with ICIs response in pan-cancer and/or a specifically dedicated sarcoma model, including SIC classification and TLS score [25,26,27], T cell inflamed signature (TIS) score [30,31,32], and immune infiltration score (IIS) [33]. Our main goal was to identify TIME recurrent patterns with potential integrative immune biomarker identification in LMS to facilitate access to LMS patients to immunotherapy clinical trial and ultimately provide a clinical tool to appropriately select LMS patients who would be better responders to ICIs.

## 2. Materials and Methods

### 2.1. Patient Samples

ICGC samples were prospectively collected as part of the International Cancer Genome Consortium (ICGC) program by the French Sarcoma Group. Clinico-pathological data and patient information are summarized in Table 1. All cases were centrally reviewed by expert pathologists of the French Sarcoma Group according to the World Health Organization guidelines and to the Fédération Nationale des Centres de Lutte Contre le Cancer (FNCLCC) grading system [1,34]. All patients provided written informed consent.

### 2.2. Whole Transcriptome Sequencing (WTS)

Total RNA from frozen primary tumor surgical samples of the ICGC cohort was extracted and sequenced using Illumina paired-end HiSeq2000 technology (Illumina Inc., San Diego, CA, USA). Detailed RNA extraction, library preparation, sequencing protocols, and data analysis were previously described [35]. TCGA RNA-seq HTSeq count data were downloaded using TCGA biolinks [36] R package (version 2.22.4) on September 2021.

In both cohorts, HTseq raw counts were transformed in Transcript Per Million using hg38 UCSC RefSeq gene models.

### 2.3. Bioinformatics and Statistical Analysis

RNAseq expression data of LMS primary tumor samples, 149 samples issued from the ICGC cohort (including multiregional tumor samples from the same tumor in 22 patients) and 74 samples from the TCGA cohort, were used for bioinformatic analysis. These related transcriptomic data matrices used two independent methodologies. The first subset used a so-called 90-gene signature method, considering the expression levels of previously characterized 93 genes related to immune checkpoint protein and membrane markers (ICP-MM) of immune cells, representative of TIME landscape [37]. This gene signature included genes considered as key markers in the immune populations such as natural killers (NKs), monocytes/macrophages, neutrophils and cytotoxic T cells, positive and negative immune checkpoint protein including the known druggable targets such as PDL1, PDL2, CTLA4, TIGIT, IDO, LAG-3, TIM-3, and key molecules involved in immune regulation. The complete list is presented in our previous publication [37]. The 3 genes KIR2DS1, KIR2DS2, and KIR2DL2 were not found in the current RNAseq expression matrix and therefore removed from the analysis.

The second method referred to as Cell-type Identification By Estimating Relative Subsets of RNA Transcripts (CIBERSORT) was a previously validated analytical tool using the expression profiles of 547 genes distinguishing a set of 22 immune cells (naïve and memory B cells, plasma cells, CD8 T cells, naïve CD4 T cells, resting memory CD4 T cells, activated memory CD4 T cells, follicular helper T cells, regulatory T cells, ᵞδ T cells, resting and activated NK cells, monocytes, macrophages (M0 macrophages, M1 macrophages, M2 macrophages), resting and activated dendritic cells, resting and activated mast cells, eosinophils and neutrophils) to derive a signature matrix that can be applied to deconvolute mixed samples in order to determine relative proportions of immune cells in TIME [38]. CIBERSORT algorithm was performed using the immunedeconv R package (version 2.0.4) and CIBERSORT R script provided by CIBERSORT authors in both relative and absolute modes. CIBERSORT computes a p-value for each sample to provide a statistical significance of the deconvolution across all cell types. Statistical analysis considered *p*-value < 0.05 as significant. Unsupervised hierarchical clustering and heatmap analysis were performed using Euclidean distance and the ward grouping function within R-bioconductor. For each heatmap, we determined the number of clusters empirically by looking at various clustering statistics such as average silhouette and gap statistics. In addition, CIBERSORT absolute abundance scores as well as T cell proportional scores (Treg was excluded) were calculated to generate an immune infiltration score (IIS). The M1/M2 macrophage score for each sample was derived by calculating the mean expression (RPKM) of these ten genes (CXCL11, IDO1, CCL19, CXCL9, PLA1A, LAMP3, CCR7, APOL6, CXCL10, and TNIP3).

Wilcoxon, Kruskal–Wallis, and Fisher’s exact tests were performed to analyze the association between immune clusters defined by the 90-gene signature method and CIBERSORT analysis, respectively, and previously published immune signatures including SIC classification, TLS score, characterized by the expression of TLS-associated B cell specific chemokine CXCL13 [25,26,27], TIS score [30,31,32], IIS score [33], macrophage score [39], as well as clinical factors including age, sex, tumor size, location, grade, HRD score, and various LMS variables. Distant metastasis-free survival (DMFS) as well as overall survival (OS) were estimated using the Kaplan–Meier (KM) method. Subgroup comparisons were performed using log-rank tests. All statistical analyses were performed by software R v4.1.2 (R Foundation for Statistical Computing, Vienna, Austria).

### 2.4. Data Availability

ICGC RNA-seq data are available on Gene Expression Omnibus under accession GSE71121. RNA-seq raw files (FastQ) on sequence read archive under accessions: SRP057793 and SRP059588 [35]. TCGA RNA-seq data are accessible from TCGA biolinks [36,40,41].

## 3. Results

### 3.1. Patient/Tumor Characterization in the ICGC/TCGA Cohorts

The ICGC and TCGA cohorts included 111 and 74 patients, respectively. Median age at diagnosis was 64 years in the ICGC cohort and 60 years in the TCGA cohort. The proportion of females was predominant in the ICGC cohort (75.7%) but similar to males in the TCGA cohort (54.1%). Most tumors were grade 2 (34.2%) and 3 (46.8%) in the ICGC cohort and mainly grade 2 (73%) tumors were reported in the TCGA cohort. The most frequent tumor location was internal trunk in both ICGC (N = 56, 50.6%) and TCGA (N = 50, 67.6%) cohorts, then limb (N = 20, 18%) in the ICGC cohort and (N = 24, 32.4%) in the TCGA cohort. Other locations including gynecological (N = 16, 14.4%), trunk wall (N = 10, 9%), head and neck (N = 5, 4.5%), and others (N = 4, 3.6%) were only recorded in the ICGC cohort but not in the TCGA cohort. Further information regarding other tumor characteristics, treatments, and clinical outcome were missing in TCGA cohorts. In the ICGC cohort, most tumors were deep (92.8%), single (97.3%), and resected (99.1%) with either R0 (61.3%) or R1 (28.8%) resection margin. All treatment and clinical outcome information were described in Table 1. It should be noted that none of the patients in both cohorts were exposed to ICIs.

### 3.2. Unsupervised Clustering Revealed a Small Subset (~15%) of LMS with Active (“Hot”) TIME with Combined 90-Gene Signature and CIBERSORT Methods

As specified above, the ICGC cohort (N = 146 tumor samples), included 22 patients with multiple tumors (between 2 and 6; 3 samples data not available) based on the 90-gene signature method, 37.6% (N = 56) tumor samples were clustered into group I-H (“hot”), while 60.4% (N = 90) were clustered to group I-C (“cold”) (Figure 1A). By visualization of a heatmap (Figure 1A), group I-H had a significantly higher gene expression of ICP-MM of immune cells than group I-C; therefore, representing active TIME. The same cohort explored with the CIBERSORT method only included 87 tumor samples after filtering deconvolution results with significant *p*-value, 18.4% (N = 16) tumor samples were clustered into group I-Hi (“High”), 19.5% (N = 17) in group I-M (“Medium”), and 62.1% (N = 54) in group I-L (“Low”) (Figure 1B). The I-H cluster was enriched with NK activated cells, M1 Macrophage, CD8 T cells, T follicular helper, plasma cells, and B memory cells, all of which are representative of active TIME and associated with ICIs response based on our knowledge to date. To note, 87.5% (N = 14/16) of tumor samples in I-H cluster identified by the 90-gene signature method significantly overlapped with group I-Hi cluster identified by CIBERSORT method (Figure 1C, *p* = 0.02), compared with smaller proportion in I-M (47%; N = 8/17) and I-L (55%; N = 27/54) clusters, showing a partial concordance of these two active immune signatures (group I-H and I-Hi) derived from two independent methodologies. From this analysis, 16.1% (N = 14/87) tumor samples had active (“hot”) TIME based on a combination of two methods.

The TCGA cohort showed similar results (N = 74). The 90-gene signature method clustered 25.7% (N = 19) tumor samples into group T-H (“hot”), which represents active TIME, and 74.3% (N = 55) into group T-C (“cold”) (Figure 2A). CIBERSORT (N = 54 after filtering deconvolution results with significant p-value) clustered 25.9% (N = 14) into group T-Hi (“high”), which represents active TIME, 20.4% (N = 11) into group T-M (“medium”) and 53.7% (N = 29) into group T-L (“low”) (Figure 2B). The majority of tumor samples in group T-Hi (78.6%, N = 11/14) compared with the smaller proportion in group T-M (9%; N = 1/11) and T-L (20.7%; N = 6/29), significantly overlapped with T-H cluster (Figure 2C, *p* = 0.0001). This overlapping analysis revealed that 11/74 (14.8%) tumor samples had active (“hot”) TIME based on both methods.

### 3.3. Immune Clusters Identified through 90-Gene Signature and CIBERSORT Methods Were Associated with Other Immune Signatures

In the ICGC cohort, group I-H and I-Hi clusters, considered as active TIME, were significantly associated with TLS score (both *p* < 0.001)), TIS score (both *p* < 0.001), IIS score (*p* < 0.001; *p* = 0.0025, respectively), high PD-L1 level (*p* = 0.018 and *p* < 0.001, respectively), as well as high macrophage score (*p* < 0.001) (Figure 3A–J). Similarly, group T-H and T-Hi clusters, representing active TIME in the TCGA cohort, were significantly associated with TLS score (*p* < 0.001 and *p* = 0.0019, respectively), TIS score (*p* < 0.001 and *p* = 0.00072, respectively), and IIS score (*p* < 0.001 and *p* = 0.00086, respectively) and high macrophage score (*p* < 0.001 and *p* = 0.0011, respectively) (Figure 4A–J).

In the TCGA cohort, group T-H cluster, but not group T-Hi cluster, was statistically associated with PD-L1 level (*p* = 0.0034 and *p* = 0.18, respectively) (Figure 4D,I). In the TCGA cohort, additional data regarding TMB, MSI, and SIC classification were available [25]. To note, the TCGA cohort showed one case with high TMB and one case with MSI in group T-H and T-Hi clusters (Figure 2A,B). Furthermore, the majority of tumor samples in the T-Hi cluster were either sub-classified in SIC E (56%) or SIC D (33%), each considered as immune high classes [25]. This contrasts with the 0% SIC E and 19% SIC D in the T-C cluster (Appendix A). Similarly, 43% SIC E was in T-Hi cluster versus 0% in T-M cluster and 14% in T-L cluster (Appendix A). All these data consistently and strongly support that group T-H and T-Hi clusters derived from two independent methods reflect an active TIME.

### 3.4. Intra-Tumor Homogeneity of TIME Signatures

We explored intra-tumor heterogeneity in terms of immune signature. In the ICGC cohort, 22 patients had multiple regions sampled and sequenced within the same tumor (2 tumor samples, N = 11; three tumor samples, N = 8; four tumor samples, N = 2, six tumor samples, N = 1). While the majority of multiregional tumor samples from the same tumor showed similar immune signature, as reflected by their close distance, some heterogeneity was observed in 4 out of 22 patients (Appendix A). For example, patient LMS 103 (blue color) had two tumor samples in the same immune cluster I-H but relatively distant from each other. Patient LMS 9 (orange color) had two tumor samples in distinct immune clusters. Patient LMS 102 (purple color) has three tumor samples (R1/2/3). R1 and R3 were in same immune cluster I-H but fairly distant from each other, whereas R2 was in distinct immune cluster I-C. LMS 18 patient (gray color) had three tumor samples (R1/2/3), including two (R1 and R2) in the same immune cluster I-H and close together but the third sample (R3) located slightly distant from these first two despite location in the same immune cluster I-H.

### 3.5. Correlation of Immune Clusters with Clinical Factors, Molecular Classifiers and Survival

There was no significant association between immune clusters (I-H and T-H) defined by the 90-gene signature method and clinical factors, sites of metastasis or LMS already known classifiers in both ICGC and TCGA cohorts (Appendix A). However, we report association between immune clusters defined by CIBERSORT method and clinical factors. For instance, the immune cluster T-Hi seemed to associate with iLMS in the TCGA cohort (Appendix A). There was no significant association between immune clusters and HRD score in the TCGA cohort (Appendix A).

Lastly but importantly, we examined if immune clusters can predict survival in ICI-naïve LMS patients. No statistical significance between immune clusters and survival outcomes (DMFS and OS) regardless of the method used (90-gene signature and CIBERSORT) in both ICGC and TCGA cohorts (Appendix A).

## 4. Discussion

Through comprehensive immune profiling in two independent and relatively large sample size cohorts, related to a single histology sarcoma type, we showed that a subset of LMS patients had active TIME, consistently associating with previously published immune signatures related to ICIs response, and which have been validated in other cancer types. The current methods support further use of multigene immune signature to improve and refine immune biomarkers to select LMS patients potentially better responders to ICIs.

PDL1 is the most studied predictive biomarker for ICIs in a variety of major cancer types including lung, breast, and gastroesophageal cancers but appeared not to predict survival or response to ICIs in LMS [14,42]. Our results showed inconsistent association between immune hot or high clusters, and high PDL1 level, which may suggest that PDL1 level alone may not be a suitable predictive biomarker for ICIs in LMS, with the caveat that only PDL1 RNA levels were evaluated in our study. Other druggable immune checkpoint proteins such as CTLA4, TIGIT, IDO, LAG-3, and TIM-3, which are more frequently expressed in karyotypically complex sarcomas such as LMS than in their counterpart (karyotypically simple sarcomas), may be of relevant use; however, the prognostic and predictive value of these biomarkers have not been successfully demonstrated in sarcoma so far [43].

The presence of tumor infiltrating lymphocytes (TILs), in particular CD8+ cytotoxic T cells, has been shown to correlate with better survival, and more importantly, the likelihood of response to ICIs in many tumor types, including STS due to their roles in an active TIME. On the contrary, T regulatory cells (Treg) represent a suppressive TIME and correlate with poor survival and resistance to ICIs [44]. The immune hot/high clusters identified in our study (I-H/I-Hi in the ICGC cohort and TCGA cohort) enriched with CD8 + T cells are correlated with high IIS score indicative of active T cell infiltration. However, no studies to date have reported a prognostic or predictive value of T cell infiltration as a sole biomarker in LMS [24].

Similarly to other karyotypically complex sarcomas, LMS are heavily infiltrated by tumor-associated macrophages, more abundant than TILs [20]. They can be polarized to classic macrophages (M1 expressing CD163+) and promote inflammatory TIME, or to M2 expressing CD68+ macrophages contributing to immune escape and considered as immune suppressive [45]. M2 phenotypes are frequently found in LMS and high level of M2 or M2/M1 ratio has been associated with worse clinical outcome in LMS, but not consistently in sarcoma in general [20,24,46]. More importantly, macrophage score (M1/M2) was recently reported to independently predict ICIs response [39]. Our results showed hot/high immune clusters (I-H and I-Hi in the ICGC cohort, and T-H and T-Hi in the TCGA cohort) defined by two independent methods (90-gene signature and CIBERSORT methods) associated with high macrophage score, suggesting that the current methods can be useful to identify a subset of LMS patients who can potentially benefit from ICIs and warrant further development.

The potential role of B cells as prognostic and predictive biomarker for ICIs in sarcoma has been recently evidenced by Petitprez et al. [25]. Using a transcriptomic analysis of TIME cell population, measuring the expression of eight immune and two stromal cell populations, the authors categorized STSs into five distinct sarcoma immune classes (SIC) including TIME immune low (Classes A and B), highly vascularized (Class C), and immune high (Classes D and E). The most inflamed SIC E immune class enriched with cytotoxic T cell and B cell lineage signatures (and characterized by the presence of TLS) was associated with a high response to pembrolizumab across all sarcoma subtypes [25]. It should be underlined that TLS was almost absent in the LMS samples [25]. However, our study demonstrated that the immune hot clusters (I-H and T-H in the ICGC and TCGA cohort, respectively) and the immune high clusters (I-Hi in the ICGC cohort and T-Hi in TCGA cohort, respectively) were significantly associated with the most highly inflamed SIC E immune class (Appendix A), and with the TLS score, reflecting the expression of TLS-associated B cell specific chemokine CXCL13 (Figure 3A,F and Figure 4A,F). The results suggest that these current methods may improve sensitivity in the detection of active TIME compared with TLS alone. To note, the presence of TLS is currently used as a stratifying biomarker in several ongoing clinical trials (NCT02406781; NCT04095208). Standardized methods to assess TLS positivity need to be further specified, and positivity is currently entrusted to well-trained sarcoma pathologist experts; as previously reported with many other immunohistochemistry biomarkers such as Ki67, inter- and intra-observer variability cannot be excluded. In addition, TLS may not be a sufficient immune biomarker to identify LMS patients who may benefit from ICIs. The PembroSarc study showed only one responder to pembrolizumab out of the six LMS patients for whom TLS-positive tumors have been identified [26]. Therefore, the predictive value of multigene signatures in patients with TLS-positive tumors need to be further investigated.

Multi-gene expression signatures reflect the overall TIME, containing a large variety of immune cell types including T cells, B cells, and macrophages. The robustness of the multi-gene expression approach is demonstrated by the consistency and significant association in the classification of immune high subgroups, such as SIC in sarcoma [25] as well as in other tumor types. Another example is the tumor inflammation signature (TIS) score, quantifying 18 specific T cell and interferon gamma (IFNᵞ) signaling pathways-related genes with key roles in coordinating and orchestrating an active but suppressive adaptive immune response in TIME [30,31,32]. TIS has been validated as predictive factor for clinical benefit to ICIs in various solid malignancies such as melanoma, head and neck cancer, gastrointestinal cancer, ovarian and triple negative breast cancer, but only in sarcoma so far [30,31,32]. Our results show that immune hot clusters (group I-H in the ICGC and TCGA cohort) and the immune high clusters (group I-Hi in the ICGC cohort and TCGA cohort) were significantly associated with TIS score and contribute to further support the potential predictive value of the methods used in this study.

It is notable that 39 genes are common in the methods currently used (90-gene signature; 547 gene CIBERSORT). The most appropriate gene sets should be further refined to best differentiate “hot” versus “cold” TIME in LMS. Further focus on the genes amenable to testing approaches in FFPE tissue would contribute to an easier translation into clinical practice, such as Nanostring and support clinical utility of these multi-gene signature assays. In addition, it is essential to validate these multi-gene signature in a clinical setting.

It is interesting to note that the multi-gene immune signatures included only a minority of overlapping genes compared with other studies. For example, only four genes are common (CD3e, granzyme, LAG3, and IDO1) between the TIS score and 90-gene signature method. In addition, few overlapping genes were used in SIC classification and the two current methods. Furthermore, *CXCL13* used for the TLS score was exclusively included in CIBERSORT method, not in the 90-gene signature. Despite few overlapping genes, a significant association of these immune signatures is reported, and hints that TIME may be identified as active using different methodologies selecting distinct sets of genes involved in immune regulation, but ultimately merging into a common biological feature, and an activated but suppressive adaptive TIME can be modulated by ICIs. Interestingly, different assays involving non-overlapping genes can have similar predictive values represent a potent tool in a single tumor type, which is not rare in oncology. For example, the three validated prognostic and predictive RNA-based multigene assays including OncotypeDx (21 gene assay), Mammoprint (70 gene assay), and Prosigna (50 gene assay) are used in the clinic to predict the efficacy of adjuvant chemotherapy in early stage hormone receptor-positive HER2-negative breast cancer [47]. However, the sensitivity, specificity, and concordance in immune signatures to characterize TIME and predict response to ICIs in a particular tumor type (i.e., LMS) requires further clinical validation.

The only immune biomarker that can potentially predict better overall survival to date in sarcoma is the presence of TLS (as B cell lineage signature), initially discovered in sarcoma, and subsequently validated in pan-tumor model. However, its use was only reported in patients treated with ICIs [25,26,27]. Since none of the patients in ICGC and TCGA cohorts were treated with ICIs, it may not be a surprise that immune cluster did not predict survival in LMS ICIs-naïve patients (Appendix A). The heterogeneity of disease characteristics and treatments may also affect survival outcome. Therefore, it appears critical to further investigate the prognostic value of such immune clusters in LMS patients treated with ICIs in the near future.

Despite the intra-tumor homogeneity of the immune signature observed in the majority of the cases, some intra-tumor heterogeneity was noted. We cannot assert, however, that the heterogeneity observed reflected different underlying tumor biology or resulted from technical and preanalytical variability. Additionally, whether immune signatures change across disease course and at recurrence or metastasis is an open issue. With only three paired primary and metastatic tumor samples from the ICGC cohort, the sample size limited any further analysis.

The main limitation of our study is that we are unable to validate the predictive value of immune clusters generated from multi-gene signature methods in a cohort of LMS patients previously treated with ICIs. Such investigation will be conducted prospectively in a forthcoming study. Interestingly, our recently published study showed that LMS responder to ICI with stable disease for over 10 months had the highest IIS score [33], whereas LMS non responder to ICI had relatively low IIS score based on CIBERSORT analysis [33]. In parallel to further characterization of the immune clusters for these two patients, a large LMS cohort is needed to further validate the predictive value of these multi-gene immune signature approaches.

## 5. Conclusions

To date, to overcome the lack of clinical predictive immune biomarkers, and considering that PDL1, TMB, and MSI are not useful in sarcoma, and in particular LMS, the promising predictive immune biomarker of TLS currently under clinical investigation requires expertise from sarcoma pathologists and inter- and intra-observer variability may not be precluded. Novel predictive immune biomarkers in LMS are therefore highly required. Our study demonstrated that RNAseq-based transcriptomic profiling is useful to identify a subset of LMS with active TIME. With further validation in a clinical cohort of patients treated with ICIs, such multi-gene immune signature approaches may help to refine current immune biomarkers to select subset of LMS patients who may benefit from ICIs in clinical trials.

## Figures and Tables

**Figure 1 cancers-15-03705-f001:**
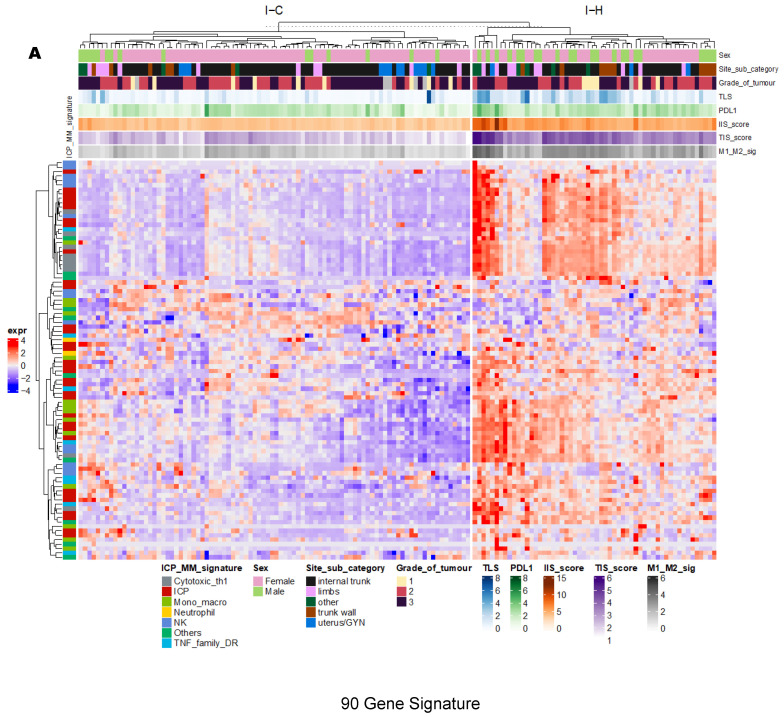
In ICGC cohort, unsupervised hierarchical clustering and heatmap analysis revealed ICGI hot (I-H) and ICGC cold (I-C) clusters using the 90-gene signature method (**A**) and ICGC-immune high (I-Hi), ICGC-immune medium (I-M), and ICGC-immune low (I-L) clusters using the Cell-type Identification by Estimating Relative Subsets of RNA Transcripts (CIBERSORT) method (**B**). I-H cluster had significantly higher gene expression of immune checkpoint protein and membrane markers (ICP-MM) and I-Hi cluster enriched with active immune cells such as CD8 T cells, Natural killer (NK) active cells, B naïve and memory cells and M1 macrophages, both of which reflected “hot” tumor immune microenvironment (TIME). Polygon graph demonstrates significant overlap between these two “hot” TIME clusters (**C**). TLS: tertiary lymphoid structure; PD-L1: program death receptor L1; IIS: immune infiltration score; TIS: T cell inflamed signature score; M1-M2_sig: macrophage score; TMB: tumor mutational burden; MSI: microsatellite instability.

**Figure 2 cancers-15-03705-f002:**
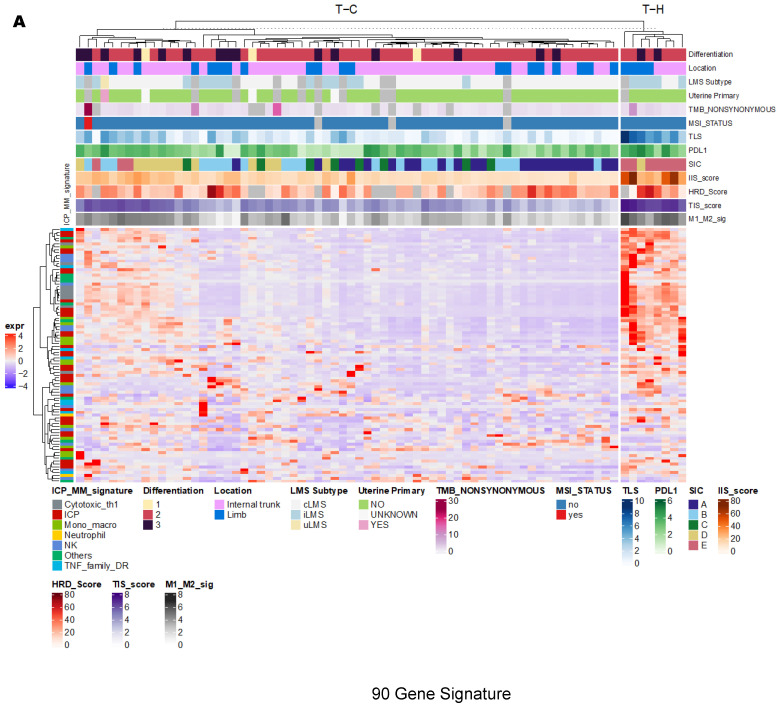
In TCGA cohort, unsupervised hierarchical clustering and heatmap analysis revealed TCGA-hot (T-H) and TCGA-cold (T-C) clusters using the 90-gene signature method (**A**) and TCGA-immune high (T-Hi), TCGA-immune medium (T-M) and TCGA-immune low (T-L) clusters using the Cell-type Identification by Estimating Relative Subsets of RNA Transcripts (CIBERSORT) method (**B**). T-H cluster had significantly higher gene expression of immune checkpoint protein and membrane markers (ICP-MM) and T-Hi cluster enriched with active immune cells such as CD8 T cells, Natural killer (NK) active cells, B naïve and memory cells and M1 macrophages, both of which reflected “hot” tumor immune microenvironment (TIME). Polygon graph demonstrates significant overlap between these two “hot” TIME clusters (**C**). ICP-MM: immune checkpoint protein and membrane markers; LMS: leiomyosarcoma; cLMS: conventional LMS; iLMS: inflammatory LMS; uLMS: uterogenic LMS; TLS: tertiary lymphoid structure; PD-L1: program death receptor L1; SIC: sarcoma immune class; IIS: immune infiltration score; HRD: homologous recombination deficiency; TIS: T cell inflamed signature score; M1-M2_sig: macrophage score; TMB: tumor mutational burden; MSI: microsatellite instability.

**Figure 3 cancers-15-03705-f003:**
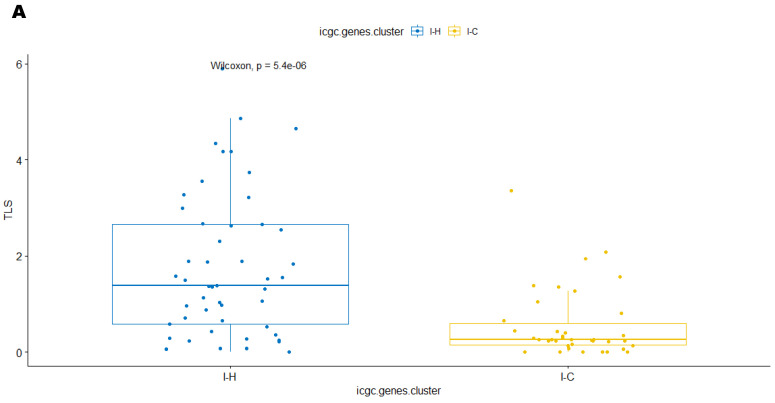
In ICGC cohort, immune “hot” clusters (I-H (identified by 90-gene signature method) and I-Hi (identified by CIBERSORT method)), were strongly associated with high TLS score (**A**,**F**), TIS score (**B**,**G**), IIS score (**C**,**H**), PDL1 level (**D**,**I**), and macrophage score (**E**,**J**). TLS: tertiary lymphoid structure; TIS: T cell inflamed signature score; IIS: immune infiltration score; PD-L1: program death receptor L1; M1-M2_sig: macrophage score.

**Figure 4 cancers-15-03705-f004:**
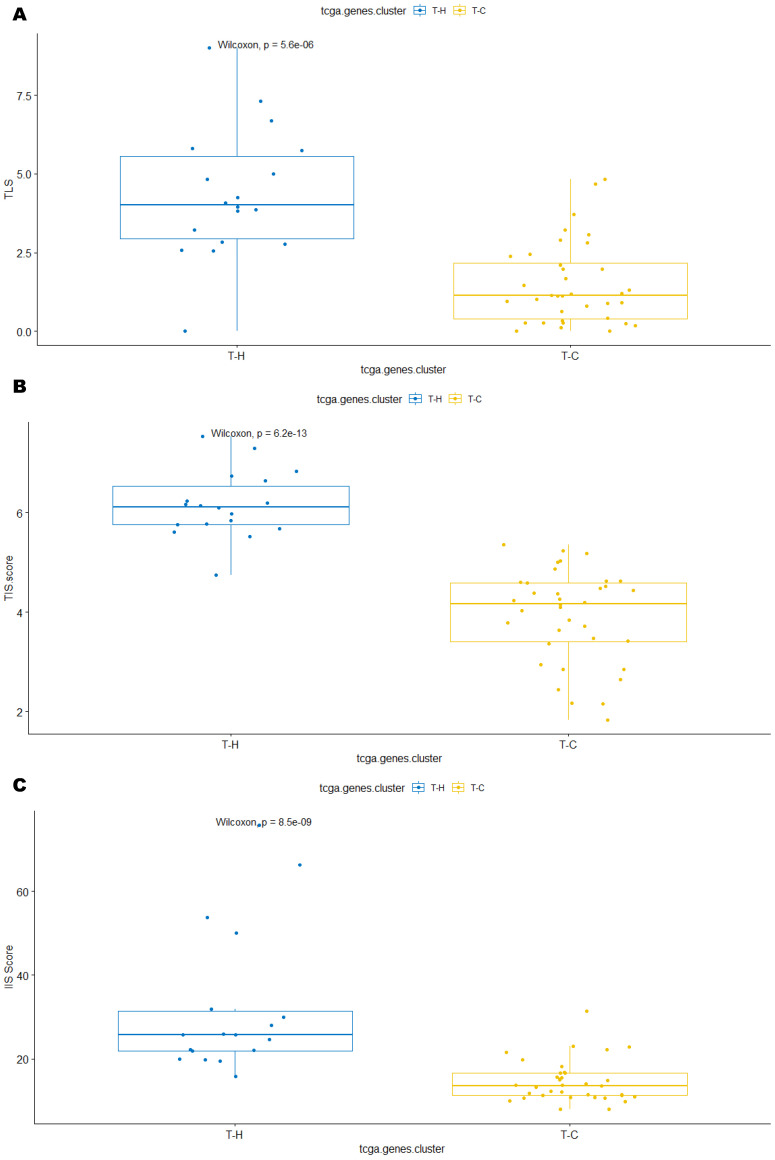
In TCGA cohort, immune “hot” clusters (T-H (identified by 90-gene signature method) and T-Hi (identified by CIBERSORT method)) were strongly associated with high TLS score (**A**,**F**), TIS score (**B**,**G**), IIS score (**C**,**H**), PDL1 level (**D**,**I**), and macrophage score (**E**,**J**). TLS: tertiary lymphoid structure; TIS: T cell inflamed signature score; IIS: immune infiltration score; PD-L1: program death receptor L1; M1-M2_sig: macrophage score.

**Table 1 cancers-15-03705-t001:** Patient and tumor characteristics were described in ICGC (N = 111 patients) and TCGA cohorts (N = 74 patients). Please note that the denominator for percentage calculation for last 3 roles, first metastatic site, locoregional treatment for metastatic disease, and systemic treatment for metastatic disease is the numbers of patients with metastatic disease (N = 62), not the numbers of whole cohort (N = 111) in ICGC cohort. * represents that 3 patients had neoadjuvant chemotherapy, 16 patients had adjuvant chemotherapy, 3 patients had chemotherapy with palliative intent due to the extent of primary disease. ** represents that 2 patients had neoadjuvant radiotherapy. *** represents that 13 patients had debulking surgery, 3 patients had cryotherapy, 5 patients had radiofrequency treatment, 9 patients had radiotherapy either alone or combined with any of aforementioned treatments.

Patient/Tumor Characteristics	ICGC Cohort (N = 111)	TCGA Cohort (N = 74)
Age		
Median (min–max)	64 (22–85)	60 (33–90)
Gender		
Female	84 (75.7%)	40 (54.1%)
Male	27 (24.3%)	34 (45.9%)
Tumor Size (cm)		Not available
≤5	27 (24.3%)	
>5 and ≤10	43 (38.7%)	
>10	40 (36%)	
Unknown	1 (0.9%)	
Tumor Grade (/FRSCC)		
1	14 (12.6%)	3 (4.1%)
2	38 (34.2%)	54 (73%)
3	52 (46.8%)	17 (23%)
Unknown	7 (6.3%)	0 (0%)
Tumor Location		24 (32.4%) 50 (67.6%)
Limb	20 (18%)	
Internal trunk	56 (50.5%)	
Trunk wall	10 (9%)	
Head and neck	5 (4.5%)	
Gynecological	16 (14.4%)	
Others	4 (3.6%)	
Tumor Depth		Not available
Superficial	8 (7.2%)	
Deep	103 (92.8%)	
Tumor Multifocality		Not available
No	108 (97.3%)	
Yes	3 (2.7%)	
Surgery		Not available
Yes	110 (99.1%)	
NA	1 (0.9%)	
Re-resection		Not available
No	96 (86.5%)	
Yes	9 (8.1%)	
Unknown	6 (5.4%)	
Surgical Margin		Not available
R0	68 (61.3%)	
R1	32 (28.8%)	
R2	1 (0.9%)	
Not evaluable/Unknown	10 (9%)	
(Neo)Adjuvant chemotherapy		Not available
No	92 (82.9%)	
Yes *	19 (17.1%) *	
(Neo)Adjuvant Radiotherapy		Not available
No	73 (65.8%)	
Yes **	38 (34.2%) **	
Local Recurrence		
No	99 (89.2%)	
Yes	12 (10.8%)	
Metastatic Recurrence		
No	49 (44.1%)	40 (54.15)
Yes	62 (55.5%)	34 (45.9%)
Metastatic Site (first)		Not available
Lung only	27 (43.5%)	
Liver only	10 (16.1%)	
Lung, liver, and others	7 (11.3%)	
Peritoneum	5 (8.1%)	
Bone	3 (4.8%)	
Skin/Soft tissue/Lymph node	9 (14.5%)	
Brain only	1 (1.6%)	
Locoregional treatment for metastatic disease		Not available
No	41 (66.1%)	
Yes ***	21 (33.9%) ***	

## Data Availability

See Methods section.

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
