# Peer review of "Comprehensive Immune Profiling Unveils a Subset of Leiomyosarcoma with “Hot” Tumor Immune Microenvironment"

_cancers, 2023, doi:10.3390/cancers15143705_

Round 1

Reviewer 1 Report

The authors present a nice study looking at a well identified subset of leiomyosarcoma patients. They employ two different transcriptomics methods to identify immune hot/active tumours across two independent data sets and compare the overlap in their identification of patient tumour profiles. They show correlation of common immune biomarkers between hot and cold identified tumours in the two methods suggesting that both have merit and that a subset of LMS patients exists that could benefit form ICI therapy, which is clinically relevant information.

Overall, the data analysis is reasonable, with appropriate methods. The conclusions drawn are not overly strong, limiting (along with other identified issues) the impact of the results somewhat. The data is presented in a simplistic fashion, with little higher order analysis, which could help derived deeper understanding. 

The authors do identify that the two methods are not entirely concordant in classifying tumours and this likely comes from the later stated caveat that the genes used to make up these profiles are not always highly overlapping (only 39 of the 90-gene and 547-CIBERSORT signatures overlap, only 4 genes between TIS score and 90-gene signature, lack of CXCL13 in the 90-gene signature). This is likely an issue  affecting many such analyses.

Are the authors able to make some comparison between the two signatures, to identify which best predicts a hot/active TME or do both signatures need to be combined to best capture all relevant patients? What is the relative predicted power of one method versus the other or of the combination of both methods? Noting there is some overlap between T-M and T-L and the T-Hot cluster (~30%).

Some editing of english language would be appropriate for clarity of particular statements but overall the language was ok.

Author Response

Thanks very much for the comments about our manuscript and inquiries

The objective of our study is not to compare these two signatures but extensively explore the tumor immune microenvironment (TIME) in LMS to identify biomarkers that could be further developed/refined to help guide ICIs treatments. Our results identified a subset of LMS having a “hot”/ adaptive TIME despite that LMS is generally thought to be “cold” tumor. Our data establishes ground work for future exploration of these two signatures --one or the other, or the combination of both in a cohort of LMS patients treated with ICIs. Unfortunately, our data could not demonstrate the relative predicted power of the one method versus the other or of the combination of both methods. This endeavor is planned in our future studies.

The second inquiry is about overlap between different clusters in TCGA cohort. The definition of T-H, T-M and T-L is based on the relative expression levels of genes included in those two signatures. For example, T-H cluster has the relatively highest expression of active tumor infiltrative immune cells such as CD8 T cells, activated NK cells, B naïve cells, Macrophages M1 and etc, but the lowest expression of inhibitory tumor infiltrative immune cells such as regulatory T cells (Tregs), resting T cells, Macrophages M2, and etc. T-L has the opposite patterns of gene expression while T-M is somewhere in between. Same principle applies to ICGC cohort. We hope this clarifies this inquiry.

Reviewer 2 Report

well written paper.  As the authors point out, the signatures that have been explored in this paper are not tested against IO therapy, and thus at best serve to generate hypothesis.

p23 line 519: correlation of DFS and OS with immune signature is likely to be weakened by the heterogeneity of prognosis and standard treatment for this group of sarcoma patients in the two databases.  This should be acknowledged.  In many tumor types with more standardized treatment, high immune signature is frequently associated with better clinical outcome even when the therapy is not immunologically based.

The high female representation in the French archive is not explained by a high level of gynecologic LMS.  Are there any explanations?

Author Response

Thanks very much for the positive comments about our manuscript.

We completely agree with the reviewer that survival data is likely affected by the heterogeneity of the prognosis and treatment received. We really appreciate this suggestion and thus modified our discussion (Page 23)

We also completely agree with the reviewer that high immune signature is frequently associated with better survival regardless of the treatments received as they are generally also a prognostic biomarker. For example, higher levels of TILs in many other tumor types such as triple negative breast cancer predicts better overall survival. Another example is high levels of PDL1 in lung cancer. To this date, these biomarkers have not shown to be consistent prognostic or predictive in LMS or sarcoma in general, as you know.

Thanks for noting that 75.5% of cohort are female but only 14.4% are gynecological LMS. We do not have a good explanation. This may be just a coincidence.

Reviewer 3 Report

In this study, the authors used bulk tumor RNA-sequencing datasets from ICGC and TCGA to characterize the tumor immune microenvironment (TIME) of leiomyosarcoma. It was reassuring to see that various bulk tumor transcriptional immune signatures (Cibersort, the "90-gene signature, TLS, SIC, etc) tended to correlate/associated with each other, although the clinical relevance and utility of the TIME features characterized by these signatures in LMS remains unclear at this time.

Minor comments:

1. How did the authors develop the "90-gene signature"?

2. How was overall survival and distant metastasis-free survival defined?  From date of diagnosis or surgery?

3. Were the specimens analyzed pre-treated?  Did any patients have tissue samples that was treatment-naive and post-neoadjuvant chemo and/or RT?  Did the transcriptional signatures change with preoperative treatment, if there were matched cases available?

Author Response

We completely agree that the data from our study is rudimentary needing further refinement, but it serves as ground work for future development of tumor immune microenvironment (TIME) biomarker in LMS that could be further validated as a clinical tool in selecting right LMS patient for ICIs.

  1. The 90 gene signature is developed by authors from their previous work cited in ref 38 Dufresne A et al. Those genes were arbitrarily selected based on their biological roles involved in the TIME from literature search and knowledge known thus far and explored/validated in ref 38.
  2. Overall survival is defined as the date of initial diagnosis to death or censored by last date of follow up. Distant disease free survival is defined as the date of initial diagnosis to the date of metastasis or censored by last date of follow up.
  3. All specimen analyzed are from surgery mostly from primary tumor. Yes, some of them (about 65%) are naïve, others received neoadjuvant chemotherapy or radiation treatment (about 35%). Unfortunately, we do not have matched samples from pre and post treatment. Therefore we cannot address the question if immune signature would be affected by the treatment. But this is a very good question and we suspect that this maybe the case which needs further exploration.